# BAP: Branch-Aware Parallel Execution for Faster DNN Inference on Mobile CPUs

## Abstract

The growing demand for real-time applications on edge devices underscores the need for faster inference of complex deep neural network (DNN) models. Although mobile devices increasingly incorporate specialized processors like GPUs and TPUs, modern DNN models such as Whisper and Vision Transformers often involve dynamic control flows and tensor operations that are incompatible and unsupported on current frameworks with these mobile accelerators. CPU presents the most viable option to improve inference latency on mobile devices due to their widespread availability, substantial memory caches, and ability to support all types of tensor operations. However, existing CPU optimization techniques focus on sequential execution, overlooking potential parallelization within Automatic Speech Recognition (ASR) and transformer-based models, leading to inefficiencies. This work introduces a novel runtime model analysis pipeline that extracts layer and branch structures from DNN model graphs to identify parallelizable branches. We propose BAP, a branch-aware memory allocation strategy that isolates memory arenas for parallel branches, reducing contention and optimizing memory reuse within each branch. Additionally, we leverage CPU multithreading to execute these branches concurrently, optimizing thread management and memory access to minimize overhead. Evaluated on ASR models and transformer-based models, our approach reduces inference latency by up to 38.5%, decreases memory allocation requirements by up to 15.6× and saves up to 20.2% energy cost compared to the TFLite naive memory allocation.

## 1 Introduction

The growing demand for real-time machine learning (ML) applications, such as voice assistants, live translation, and augmented reality, has intensified the need for faster deep learning inference on edge devices like smartphones and single-board computers (Yang et al., 2021; Dong et al., 2022; Xu et al., 2023; Rekesh et al., 2023). On-device inference offers significant benefits over cloud-based alternatives, eliminating the need to send data to external servers, which introduces privacy risks and delays. To address the demands of efficient edge inference, frameworks such as TensorFlow Lite (TFLite) have been widely adopted (Abadi et al., 2015). TFLite, for instance, employs quantization and operator fusion to optimize model size and computational efficiency (Nguyen et al., 2020; Orăşan et al., 2022; Xu et al., 2021; Prasad et al., 2020), while leveraging hardware accelerators such as GPUs, TPUs, and NPUs for static models (Park & Kim, 2023; Lee et al., 2019; Xu et al., 2022). Similarly, MNN enhances inference speed by optimizing kernel execution and aggressively reusing memory (Jiang et al., 2020). These frameworks have achieved notable success in tasks such as image classification (e.g., MobileNet) and object detection (e.g., YOLO), where model structures are static and predictable. Beyond these frameworks, many studies have further optimized edge inference. Techniques such as co-execution of CPU and GPU tasks, memory-efficient transformations, and dynamic model partitioning across heterogeneous processors have made strides toward more efficient inference (Kim et al., 2019; Jeong et al., 2022; Jia et al., 2022; Wang et al., 2018; Wei et al., 2023). However, despite these optimizations, significant challenges arise when deploying models with dynamic control flows and tensor operations, such as ASR models and transformer-based architectures. These models are incompatible with hardware accelerators optimized for static workloads on existing mobile inference frameworks, making traditional optimization techniques ineffective (Cordesius et al., 2021). Furthermore, existing solutions often require extensive model refactoring or retraining,

which is impractical for large pre-trained models and can degrade performance during deployment (Kim et al., 2019; Wei et al., 2023). This highlights the need for new approaches that can handle the unique challenges posed by these models on off-the-shelf edge devices.

CPUs remain the most practical option for executing ML models on edge devices due to their flexibility in handling dynamic workloads and widespread availability (Zeng et al., 2023). Studies show minimal performance differences between CPUs and GPUs on mobile devices (Zhang et al., 2023; Wei et al., 2023; Bordawekar et al., 2010), reinforcing the viability of CPUs in this setting. ASR and transformer-based models also bring additional advantages when paired with CPUs. These models naturally lend themselves to parallel execution due to internal structures like multi-head attention (Cao et al., 2012; Gulati et al., 2020; Vaswani, 2017). This built-in balance across different computational branches reduces the synchronization overhead typically associated with parallel processing. CPUs, with their robust multithreading capabilities, are well-equipped to take advantage of this structural balance, distributing tasks efficiently across cores. This not only enhances the performance of parallel tasks but also allows CPUs to manage both parallel and sequential operations without the need for significant model modifications. Nevertheless, fully leveraging CPU-based parallelism presents several challenges: (1) accurately analyzing model graphs to identify operations and branches that can be executed in parallel; (2) managing memory allocation in a way that avoids data dependency conflicts, such as Read-After-Write (RAW) dependencies, where one operation might overwrite data that another operation still needs; (3) minimizing the overhead of parallel execution, as unbalanced workloads or inefficient thread synchronization can negate the benefits of parallelism; and (4) avoiding significant model refactoring, as this adds complexity and could hinder the deployment process.

To address these challenges, we propose **BAP**, a novel optimization approach for ASR and transformer models on mobile CPUs. BAP dynamically identifies parallelizable branches in model computational graphs via a runtime analysis pipeline. We introduce a branch-aware memory allocation strategy that isolates memory arenas for parallel branches, reducing data conflicts and optimizing cache locality. Leveraging CPU multithreading with optimized thread management and memory access, our method ensures efficient parallel execution without modifying model structures, meeting the critical need for real-time performance on mobile devices. Evaluated on devices like Dimensity, Google Tensor, Kirin-powered Android devices, and Raspberry Pi 4B, BAP achieved a 38.5% reduction in inference latency, up to $15.6\times$ memory allocation improvements over TFLite's naive allocation, and up to 20.2% energy savings. Our contributions are as follows:

- We introduce BAP, a CPU-specific optimization system for ASR and transformer-based models with dynamic control flows and tensor operations. BAP combines branch-aware memory allocation and multithreading to optimize parallel execution and reduce latency.

- We develop a branch-aware memory allocation strategy that reduces data contention by isolating memory arenas for parallel branches, enabling safe and efficient parallel execution.

- BAP delivers substantial improvements in latency, memory efficiency, and energy consumption without requiring model refactoring, supporting practical real-time inference on edge devices.

## 2 RELATED WORK

**Model Optimization Strategies**: The demand for optimizing ML inference on edge devices has led to strategies like quantization, operator fusion, and pruning (Yao et al., 2020; Kim et al., 2022; Jiang et al., 2022; Niu et al., 2021). Quantization reduces precision to shrink model size and computational cost, lowering latency and energy consumption for static models. Operator fusion combines multiple operations into a single step to minimize memory access overhead, while pruning removes less essential components to reduce computational load. However, techniques like quantization-aware training and post-training quantization often require access to validation datasets or retraining, which may not always be feasible (Nguyen et al., 2020). Additionally, these methods can reduce accuracy and depend on model-specific tuning, limiting their applicability when models must be deployed without retraining or extensive infrastructure.

Another strategy optimizes model structures by replacing computationally expensive operations with more efficient alternatives. Models like MobileNet (Howard, 2017) and MobileBERT (Sun

et al., 2020) use techniques such as depthwise separable convolutions and attention-based pruning to reduce parameters while maintaining accuracy. For example, MobileNet lowers FLOP counts by splitting convolutions into depthwise and pointwise operations, and MobileBERT simplifies BERT's architecture with minimal performance loss. Similarly, LookupFFN replaces compute-heavy matrix multiplications in transformers with memory-efficient lookup operations, improving suitability for CPU inference (Zeng et al., 2023). However, these methods require careful design to balance efficiency and accuracy, may be scenario-specific, and focus on single-threaded efficiency without fully exploring parallelism within the computation graph on edge devices.

**Hardware Acceleration and Heterogeneous Computing**: As edge devices adopt heterogeneous processors, hardware acceleration is key to improving inference speed (Symons et al., 2022). Frameworks like TFLite and MNN offload tasks to accelerators like GPUs, TPUs, and NPUs, enhancing performance for static models. Advanced techniques such as co-execution dynamically partition tasks between CPUs and GPUs for better utilization. For instance, CoDL enables intra-operator parallelism to optimize latency and energy efficiency (Jia et al., 2022), while NN-Stretch transforms sequential models into parallel branches for independent execution across processors (Wei et al., 2023). BAND manages concurrent DNN inference on heterogeneous processors (Jeong et al., 2022), and methods like uLayer (Kim et al., 2019) and OPTiC (Wang et al., 2018) enhance CPU-GPU co-execution. However, these approaches struggle with dynamic models like ASR and transformers due to inefficient offloading, leading to CPU fallback. Additionally, co-execution methods often require separate memory allocations to avoid conflicts, increasing memory usage (Wang et al., 2023). While improving hardware utilization, these methods complicate memory management and often require model modifications, hindering deployment and scalability.

**Sequential Execution and Memory Management**: SOTA frameworks like TFLite and MNN use sequential execution strategies, traversing computation graphs node by node based on data dependencies (Lee & Pisarchyk, 2020). While this simplifies memory reuse and ensures correctness, it limits parallel execution, especially in models with branching structures like ASR and transformers. These frameworks aggressively reuse memory to reduce the footprint, but this leads to data dependency conflicts that hinder parallelism. As a result, despite being optimized for memory efficiency, they fail to fully exploit the parallelism in the computation graph, resulting in suboptimal performance. To address these limitations and accelerate ASR models, we propose a CPU-based parallelism framework with branch-aware memory allocation, optimizing inference on edge devices without modifying the model.

## 3 BAP System Design

Our framework dynamically identifies parallelizable branches in DNN models through a runtime graph analysis pipeline, as shown in Figure 1. This analysis enables branch-aware memory allocation, which reduces memory conflicts and enhances execution efficiency. Additionally, efficient multithreading ensures balanced workload distribution and minimizes overhead, resulting in significant latency and allocation improvements.

### 3.1 Graph Analysis Pipeline

Identifying parallelizable branches is essential for enabling efficient parallel. The main challenge arises from the varied structures of model graphs $\mathcal{G} = (V, E)$, where $V$ is the set of nodes and $E$ is the set of edges representing data dependencies. This complexity makes it difficult to extract parallelism without disrupting the data flow. Even when branches can be executed in parallel, unbalanced workloads can reduce the benefits of parallelism, requiring careful criteria to identify balanced branches. To address this, we design a graph analysis algorithm that splits the model into layer-branch structures, isolating independent branches for parallel execution. This approach is adaptable to various models, ensuring generalizability while assessing workload balance.

### 3.1.1 Definitions: Layer and Branch

In our graph analysis, we define two core concepts that are fundamental for isolating parallelizable segments: branch and layer.

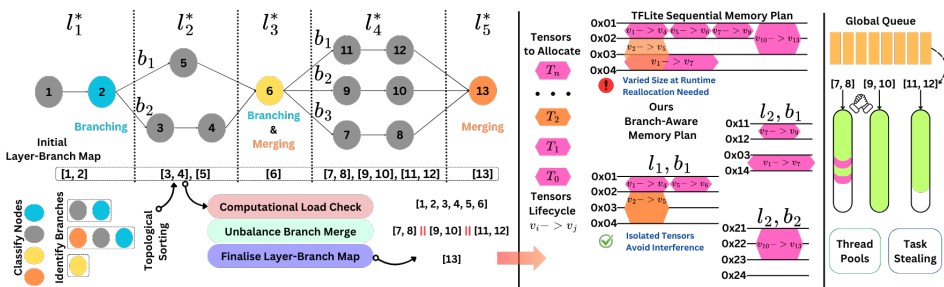

Figure 1: BAP System Overview: (1) nodes are classified into categories (sequential, branching, merging) and grouped into branches (e.g., $b_1$, $b_2$) for parallel execution, (2) a computational load check and branch merging optimize resource use, (3) tensors $T_0$, $T_1$, etc., are allocated and managed in branch-specific memory arenas to reduce conflicts due to dynamic memory reallocation needs, and (4) efficient multithreading with thread pools and task stealing ensure balanced workload distribution.

**Branch**: A branch $b$, or subgraph $\mathcal{S}$, is a set of sequentially connected nodes $\{v_1, v_2, \ldots, v_k\}$ within a layer. A subgraph may have multiple inputs or outputs, but within the subgraph, the nodes are strictly connected in a sequential manner. This ensures that each subgraph can be executed independently of others within the same layer, provided there are no inter-branch dependencies.

**Layer**: A layer $\ell$ in the computation graph is a set of branches that can be executed concurrently, with all dependencies resolved. Subgraphs in the same layer have no unmet dependencies, allowing parallel execution, while outputs from one layer feed into the next, ensuring proper data flow and execution order.

The motivation for this definition is to minimize interference between branches and layers and the rest of the graph. This structure forms the foundation for enabling parallel execution, ensuring that parallel branches do not disrupt data flow in other parts of the model.

### 3.1.2 NODE CLASSIFICATION AND SUBGRAPH PARTITIONING

Our algorithm, *Algorithm 1*, begins by classifying each node $v_i \in V$ in the graph into one of four categories: (1) Sequential nodes have a single input and output, representing linear connection; (2) Branching nodes have a single input but multiple outputs $v_i \rightarrow \{v_j, v_k, \ldots\}$, indicating points where the flow diverges into parallel paths; (3) Merging nodes have multiple inputs converging into a single output $\{v_j, v_k, \ldots\} \rightarrow v_i$, requiring synchronization of parallel paths; (4) Branching-merging nodes combine both branching and merging behaviours, representing more complex graph structures that require careful handling.

After classifying the nodes, we traverse the graph to group nodes into $\mathcal{S}$. Starting from each unvisited node, sequential nodes are added to the current $\mathcal{S}$ until a branching or merging node is encountered. When a branching node is reached, the current group is finalized, and the branching node starts a new $\mathcal{S}$, allowing independent parallel execution. Similarly, encountering a merge node finalizes the group to ensure proper synchronization. For branching-merging nodes, the current group is immediately finalized, treating the node as an isolated branch due to its complexity. After forming each $\mathcal{S}$, we eliminate duplicates by comparing their structures, ensuring only unique execution groups remain. This results in a set of independent $\mathcal{S}$s that can be executed sequentially or in parallel, depending on their structure and dependencies.

### 3.1.3 TOPOLOGICAL SORTING AND PARALLELIZABLE LAYER IDENTIFICATION

After partitioning the $\mathcal{G}$ into $\mathcal{S}$s, we perform topological sorting to establish an execution order that respects dependencies and maximizes parallel execution (*Algorithm 2*). The in-degree $d[i]$ of each $\mathcal{S}$ is calculated, where $d[i]$ represents the number of unresolved dependencies for each $\mathcal{S}$. $\mathcal{S}$ with $d[i] = 0$ are added to the execution queue $Q$, and as they are processed, the in-degrees of their dependents $d[j]$ are reduced. This process ensures that independent $\mathcal{S}$s are grouped into

**Algorithm 1** Subgraph Partitioning

1: **Input**: Graph $\mathcal{G} = (V, E)$ with nodes and connections
2: **Output**: Partitioned subgraphs $\mathcal{S}$s
3: **for** each node $v_i \in V$ **do**
4:     Count incoming and outgoing edges
5:     **if** in_count $> 1$ **and** out_count $> 1$ **then**
6:         Mark as *branching-merging*
7:     **else if** in_count $> 1$ **then**
8:         Mark as *merging*
9:     **else if** out_count $> 1$ **then**
10:         Mark as *branching*
11:     **else**
12:         Mark as *sequential*
13:     **end if**
14: **end for**
15: **for** each unvisited node $v_i \in V$ **do**
16:     **if** $v_i$ is visited **then**
17:         continue
18:     **end if**
19:     Initialize $\mathbb{S}$, add $v_i$
20:     **while** unvisited neighbors exist **do**
21:         **if** $v_i$ is *branching* or *merging* **then**
22:             Finalize $\mathcal{S}$, start new one
23:         **end if**
24:         Add $v_i$ to $\mathcal{S}$
25:     **end while**
26: **end for**
27: Remove duplicate $\mathcal{S}$s from $\mathbb{S}$

**Algorithm 2** Topological Sorting of Subgraphs

**Input**: Subgraphs $\mathbb{S} = \{\mathcal{S}_1, \mathcal{S}_2, \ldots, \mathcal{S}_n\}$ with dependencies.
2: **Output**: Layer-Branch Map $\boldsymbol{P}$ with sorted layers.
Initialize in-degree $d[i] = 0$ for all $\mathcal{S}_i$.
4: **for** each dependency $(\mathcal{S}_i, \mathcal{S}_j) \in E$ **do**    ▷ Build in-degrees
    $d[j] \leftarrow d[j] + 1$
6: **end for**
Initialize queue $\mathbb{Q}$ with all $\mathcal{S}_i$: $d[i] = 0$
8: **while** $\mathbb{Q}$ is not empty **do**
    Initialize empty current layer $\ell$
10:     **for** each subgraph $\mathcal{S}_i \in \mathbb{Q}$ **do**
        Add $s_i$ to current layer $\ell$
12:         **for** each dependent subgraph $\mathcal{S}_j$ of $\mathcal{S}_i$ **do**
            $d[j] \leftarrow d[j] - 1$
14:         **if** $d[j] = 0$ **then** ▷ Add to queue when in-degree is zero
            Add $\mathcal{S}_j$ to $\mathbb{Q}$
16:         **end if**
        **end for**
18:     **end for**
    Add layer $\ell$ to layer-branch map $\boldsymbol{P}$
20: **end while**

layers for concurrent execution. Within each layer, $\mathcal{S}$s are sorted by node index for consistency, and consecutive layers with single $\mathcal{S}$ are merged to simplify the structure. This results in a layer-branch map $\boldsymbol{P} = \{\ell_1, \ell_2, \ldots, \ell_n\}$ that organizes $\mathcal{S}$s into layers suitable for parallel or sequential execution.

For efficient parallel execution, we focus on layers with two or more branches containing enough computationally intensive nodes, like matrix multiplications, to justify the overhead of parallelism. Layers that don't meet these criteria are processed sequentially, minimizing unnecessary thread management and context switching. The resulting $\boldsymbol{P}$ forms the basis for efficient memory allocation, enabling resource reuse without conflicts.

### 3.2 BRANCH-AWARE MEMORY ALLOCATION

Efficient memory management is essential for running DNNs on edge devices. SOTA frameworks typically manage allocation via a memory arena in sequential execution, freeing intermediate tensors after use. However, parallel execution complicates this due to uncertainties in tensor lifecycles.

#### 3.2.1 ADDRESSING RAW DATA DEPENDENCIES

**Challenge**: In frameworks like TFLite, tensor lifecycles are typically tracked by identifying the first and last nodes that use a given tensor (Lee & Pisarchyk, 2020). This approach assumes a sequential execution process, where the last node in the lifecycle completes after all prior operations, allowing for safe memory reuse. However, parallel execution disrupts this assumption, as the last node may execute earlier than expected due to concurrent branch execution. This can lead to premature memory reclamation, causing RAW conflicts when other nodes in the graph still require access to the tensor.

**Proposed Solution**: To address these challenges, we propose a branch-aware memory allocation strategy that isolates memory management at the granularity of layers and branches. Each branch-specific memory arena is indexed by both layer and branch identifiers, $\mathcal{A}_{\ell_i}^{\mathcal{S}_i}$. Within each branch, we ensure memory alignment to optimize access speed.

We enhance tensor lifecycle tracking by encoding not only node-level dependencies but also the first and last layers and branches that use each tensor. Each node is assigned a unique 32-bit identifier that encodes its layer, branch, and local index (from $\boldsymbol{P}$) within the branch:

$$\text{NodeID}(i) = (\text{LayerID}(i) \ll 16) \,|\, (\text{BranchID}(i) \ll 8) \,|\, (\text{LocalIndex}(i)\&0xFF)$$

where $\ll$ denotes a bitwise left shift, $|$ denotes a bitwise OR, and $\&$ denotes a bitwise AND operation. This encoding allows for precise control over memory management across concurrent branches. Tensors are allocated in the memory arena of the branch where they are first used. Specifically, if a tensor $T_i$ is first used in branch $\mathcal{S}_i$ of layer $\ell_i$, it is allocated in that branch's memory arena: $\text{Alloc}(T_i) = \mathcal{A}_{\ell_i}^{\mathcal{S}_i}$. Input tensors are preserved throughout their lifecycle since they may be needed by multiple branches. Similarly, output tensors are protected until all relevant branches have completed processing. For intermediate tensors used exclusively within a branch, memory can be reclaimed once the operations in that branch are complete, without affecting tensors in other branches.

### 3.2.2 HANDLING DYNAMIC REALLOCATION

**Challenge**: Dynamic operations lack predefined tensor shapes until runtime, requiring temporary memory allocation followed by reallocation when actual sizes are determined. In parallel execution, this reallocation introduces synchronization bottlenecks, as threads must wait for memory availability, negating the benefits of parallelism. Additionally, reallocating memory prematurely can cause RAW conflicts by overwriting tensors still in use by other branches.

**Proposed Solution**: Our approach focuses on restricting reallocation to the current branch and its subsequent layers. When a dynamic tensor's size is determined, only the directly affected tensors undergo reallocation, following these conditions:

$$\text{Reallocate}(T_j) \quad \forall j \in \{\text{current branch} \cup \text{future layers}\} \quad \text{if size}(T_j) \neq \text{known}$$

This branch-specific reallocation prevents interference with other concurrently executing branches. Unlike conventional frameworks, where reallocations in one part of the graph can disrupt other branches, our method ensures independent branch execution. By minimizing synchronization bottlenecks, other branches can continue uninterrupted, improving parallel execution efficiency.

### 3.3 MULTITHREADING EXECUTION

To efficiently execute parallel branches, we maintain a fixed thread pool to reduce overhead from thread management. This allows threads to be dynamically assigned to tasks without additional setup costs. To optimize CPU utilization, we implement task stealing, enabling idle threads to handle tasks from busy ones, ensuring balanced workload distribution. By maintaining branch isolation and minimizing synchronization points, we reduce contention between threads, particularly during dynamic tensor reallocations. This approach maximizes parallelism while minimizing multithreading overhead across devices.

## 4 EVALUATION

### 4.1 IMPLEMENTATION

We conducted experiments using pretrained SOTA transformer and ASR models from Huggingface and GitHub without modifying architectures or weights. Parallel inference methods were integrated into the latest TFLite 2.17.0 with key runtime modifications in the *Invoke* and *InvokeImpl* functions, targeting *Subgraph* and *Interpreter* objects. These changes enabled branch-aware execution, isolating parallel branch processing from the original sequential model. We wrote $2,116$ lines of C++ code for model analysis algorithms and branch-aware memory allocation, identifying parallelizable branches and optimizing memory usage during execution. Also, custom testing tools were created to evaluate performance across platforms including Kirin, Dimensity and Google Tensor-powered Android devices, as well as Raspberry Pi 4B, measuring inference latency, memory usage and estimated energy consumption.

Table 1: Test Platforms and Model Details. Cs: Cores; $\ell$s: Layers; PAR-$\ell$s: Layers that contain multiple branches; MAX BR: Maximum branches; PARS: Parameters; PAR-Ops: Percentage of parallelized operations.

| TEST PLATFORMS | | | | MODEL DETAILS | | | | | |
|---|---|---|---|---|---|---|---|---|---|
| DEVICE | Cs | FREQ. | SOC | MODEL | $\ell$s | PAR-$\ell$s | MAX BR. | PARS | PAR-Ops |
| K50 | 8 | 2.85 GHz | Dimensity 8100 | Whisper | 123 | 88 | 8 | 912M | 61.43% |
| P30 Pro | 8 | 2.60 GHz | Kirin 980 | Conformer CTC | 37 | 18 | 3 | 67.7M | 8.46% |
| Pixel 6 | 8 | 2.80 GHz | Tensor | MobileViT-S | 19 | 9 | 3 | 130M | 17.93% |
| Pi 4B | 4 | 1.80 GHz | BCM2711 | MobileViT-XS | 19 | 9 | 3 | 106M | 17.93% |

## 4.2 EXPERIMENTAL SETUP

**Models**: We evaluated four models to assess our method's effectiveness: Whisper (INT8) and Conformer CTC (FLOAT16) for ASR, and MobileViT-S (FLOAT16) and MobileViT-XS (FLOAT16) for image recognition. These models vary significantly in size and complexity, with total layers ranging from 19 to 123 and parallelizable layers from 9 to 88. Table 1 lists the maximum number of parallelizable branches and the number of parameters (in millions) for each model, providing insight into their resource demands. Compared to models like ResNet or EfficientNet, which have 11M to 32M parameters, the larger size and complexity of ASR models highlight the need for optimization. Current solutions struggle to handle them on mobile devices, making BAP beneficial.

**Devices**: We tested these models on a diverse set of platforms, ranging from high-end smartphones to single-board computer. The selected devices include the Xiaomi K50, Huawei P30 Pro, Google Pixel 6, and Raspberry Pi 4B, all detailed in Table 1. Each device differs in CPU core count (Cs), maximum clock frequency (FREQ.), and System-on-Chip (SOC) architecture, offering a comprehensive evaluation across different hardware configurations.

**Performance Metrics**: We evaluated our method by measuring runtime inference latency, memory usage, and power consumption, averaging each over five runs. Experiments utilized all available CPU cores, and we also tested different core counts for comparison. For MobileViT-XS and MobileViT-S, we used 10 images from the TensorFlow Flowers dataset (Paul, 2023) with an input size of $224 \times 224$. For Conformer CTC and Whisper, we used five audio samples (3–10 seconds each) from the LibriSpeech dataset (Panayotov et al., 2015) at a 16 kHz sample rate. Inference latency was tracked per inference pass using on-device profilers. Memory usage was monitored by profiling peak runtime memory and measuring the memory allocation arena size. Power consumption was measured using the Android BatteryManager API (Android Developers, 2024), capturing current and voltage at 10 millisecond intervals to compute power and total energy.

**Baselines**: We compared our method against two memory plans in TFLite. First, we tested against the standard **TFLite runtime**, which uses the Arena memory plan for aggressive sequential allocation reuse, and this plan was used for comparison in terms of performance metrics beyond memory. Second, we evaluated **TFLite's naive** memory plan, which assigns separate memory to each tensor. While we compared allocation memory efficiency against both plans, we focused on comparing other aspects like latency and energy consumption against the first, as it represents TFLite's optimal performance. Also, we did not include methods like CoDL or NN-Stretch, as they mainly focus on DNNs without dynamic control flows and rely on heterogeneous processor co-execution. In this case, the TFLite runtime remains the SOTA for CPU-only execution.

## 4.3 OVERALL RESULTS

Our method preserves the model's weights and structure, ensuring that outputs and accuracy remain identical to the original pretrained model during testing. While we focus on improving inference latency, memory efficiency, and power consumption, the functional performance and accuracy remain unchanged across all evaluations.

### 4.3.1 LATENCY

BAP consistently outperforms the state-of-the-art TFLite runtime across all tested models and devices, achieving latency reductions ranging from 14% to 38%. For smaller models like MobileViT-

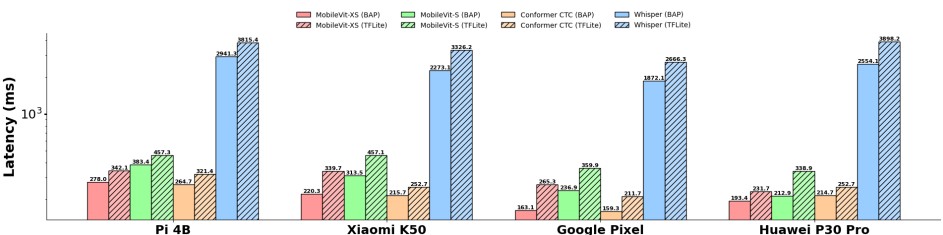

Figure 2: Latency Comparison: BAP vs TFLite runtime.

Table 2: Memory Allocation Comparison for Different Methods (in MB)

| MODEL | Ours | TFLite Naive | TFLite Runtime |
|---|---|---|---|
| Conformer CTC | 26.64 | 284.15 | 23.39 |
| Whisper | 227.27 | 3535.6 | 144.95 |
| MobileViT-S | 82.74 | 538.43 | 38.74 |
| MobileViT-XS | 77.98 | 437.76 | 31.15 |

XS, improvements are more pronounced on higher-end devices, with latency reductions of 38.5% on the Google Pixel and 35.1% on the Xiaomi K50. Even on low-powered devices like the Raspberry Pi 4B, we observed an 18.7% reduction in latency. Similarly, for MobileViT-S, latency improvements were 16.1% on the Raspberry Pi and 31.4% on the Xiaomi K50, demonstrating our method's scalability across different hardware configurations (see Figure 2). Although TFLite also utilizes multi-core processing, its multithreading support for floating-point models remains limited. Specifically, TFLite's parallelization is constrained to certain operations, leaving much of the computation serialized and diminishing its ability to fully capitalize on multi-core architectures. In contrast, our method optimizes parallel subgraph execution, leveraging all available CPU cores more effectively to achieve greater latency reductions.

**How does model complexity impact BAP?** For ASR models, Conformer CTC showed consistent but smaller improvements, with latency reductions ranging from 14.6% to 24.7%, with the highest gains on the Google Pixel. Whisper, the largest model tested, saw significant improvements due to its highly parallelizable layers, achieving a 22.9% reduction on the Raspberry Pi and up to 31.7% on the Xiaomi K50. With 88 parallelizable layers, Whisper is well-suited for branch-aware parallel execution, leading to noticeable performance gains. Our method also performs effectively on transformer-based models with dynamic operations, like MobileViT, reducing latency by up to 38.5% on high-end devices.

**How does hardware impact BAP?** Devices with more CPU cores and higher clock speeds, such as the Xiaomi K50 and Google Pixel, consistently achieved higher latency reductions, particularly for models with extensive parallelizable branches, like Whisper and MobileViT. Lower-end devices, such as the Raspberry Pi, still saw notable improvements, though their constrained hardware limits the degree to which our method can parallelize execution.

### 4.3.2 MEMORY AND ENERGY CONSUMPTION

**Peak Runtime Memory**: For peak runtime memory (Figure 3), the differences between our method and TFLite runtime remain modest across all devices, typically less than 5%. On the Raspberry Pi 4B, the increase in memory usage for models like Conformer CTC and MobileViT-S was 5.93 MB (min.) and 11.37 MB (max.), respectively. The Xiaomi K50 showed a minimum increase for MobileViT-S (0.39 MB), and Conformer CTC saw a larger rise of 19.34 MB (max.). On the Google Pixel, Whisper exhibited an increase of 7.89 MB (max.). The Huawei P30 Pro recorded a peak memory difference of 13.6 MB (max.) for Whisper. Overall, while our method introduces some runtime memory overhead, this remains minor, even for the most complex models.

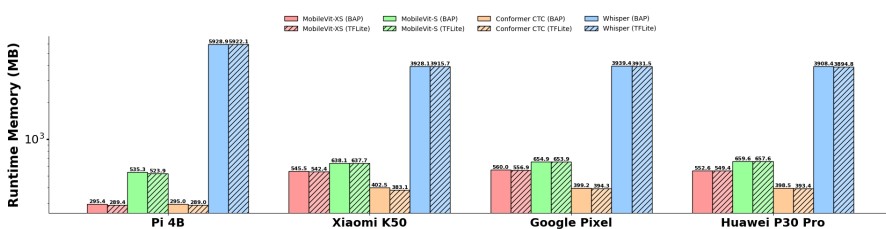

Figure 3: Peak Runtime Memory Usage Comparison: BAP vs TFLite runtime

**Allocation Memory**: Our branch-aware memory plan significantly reduced memory allocations compared to the naive strategy across various models. For Conformer CTC, it used $10.7\times$ less memory than the naive approach and was only $1.14\times$ larger than the Arena plan. Whisper showed the most drastic improvement, using $15.6\times$ less memory than naive allocation and only $1.57\times$ more than the Arena plan. For MobileViT-S and MobileViT-XS, our method used $6.5\times$ and $5.6\times$ less memory than the naive approach, respectively, and only $2.14\times$ and $2.5\times$ more than the Arena plan. Although our method requires slightly more memory than the TFLite Arena plan due to limiting tensor reuse within each branch, it enables safe and effective parallelism. Overall, our branch-aware memory allocation efficiently reduces memory usage compared to the naive approach while providing faster inference than TFLite Runtime

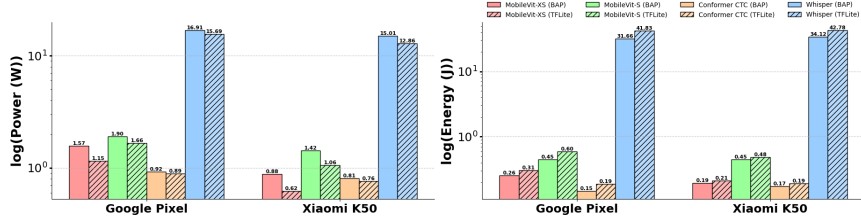

Figure 4: Power and Energy Comparison on Pixel 6 and K50: BAP vs TFLite runtime

**Energy Analysis**: Power and energy measurements on both Google Pixel and Xiaomi K50 showed that, although BAP consumed more power than TFLite, it was more energy-efficient overall due to significant reductions in inference time. On Google Pixel, BAP achieved energy savings of 16.07% to 24.64%, while on the Xiaomi K50, energy consumption decreased by 7.94% to 20.19%, with the largest gains seen for Whisper. This demonstrates that while BAP's multithreading increases CPU utilization and power draw, the improved efficiency in inference time ultimately leads to lower total energy consumption (see Figure 4). This balance between higher power use and faster execution highlights BAP's effectiveness for energy-constrained, real-time applications on mobile devices.

The increase in memory and power consumption in our method aligns with findings from CoDL (Jia et al., 2022), where higher power usage on edge devices is balanced by reduced total energy consumption. Both approaches show that while power demand rises due to parallel processing, significant reductions in inference time ultimately lead to improved energy efficiency. This trade-off is essential for real-time, performance-sensitive applications, where the modest resource overhead is outweighed by the performance gains.

### 4.3.3 PARALLELIZATION EVALUATION

**Layer parallelism Analysis**: Table 3 shows that our method significantly accelerates layers with parallelizable branches in both MobileViT-XS and Whisper models, reducing inference time by up to 67.7% in Whisper and 58.1% in MobileViT-XS. Layers with three or more parallel branches—such as Layer 2 in MobileViT-XS and Layer 1 in Whisper—benefit the most, showing substantial latency reductions. Conversely, layers without parallelizable branches, where computation remains largely sequential, experience slight latency increases due to multithreading overhead. However, this trade-

Table 3: Layer-Wise Inference Latency Comparison on Google Pixel 6 (in ms)

| MobileViT-XS | | | | Whisper | | | |
|---|---|---|---|---|---|---|---|
| Layer ID | TFLite | BAP | BR. | Layer ID | TFLite | BAP | BR. |
| 1 | 42.16 | 46.59 | 1 | 1 | 48.54 | 15.69 | 4 |
| 2 | 3.28 | 1.56 | 3 | 2 | 11.90 | 16.59 | 1 |
| 3 | 31.04 | 31.78 | 1 | 3 | 41.27 | 16.59 | 3 |
| 4 | 3.46 | 1.45 | 3 | 4 | 5.19 | 2.51 | 8 |
| 5 | 34.69 | 35.50 | 1 | 14 | 62.55 | 66.13 | 1 |
| 6 | 1.23 | 0.74 | 3 | 16 | 139.00 | 155.05 | 1 |

off is minimal, as the performance gains in parallelizable layers far outweigh the minor overhead in non-parallel layers, making our method highly effective in accelerating complex ASR inference.

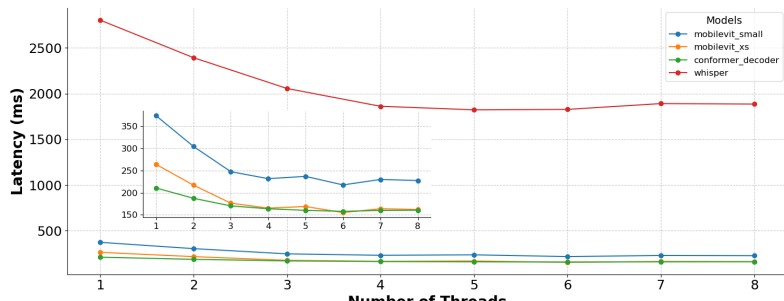

Figure 5: Thread Count Impact on Google Pixel

**Thread Count Impact**: Figure 5 shows that increasing threads significantly reduces inference time for all models up to 3 threads, which matches the maximum parallelizable branches for Conformer CTC and MobileViT. For example, MobileViT-S drops from 373.6 ms to 247.4 ms, and Conformer CTC from 210.6 ms to 170.1 ms with 3 threads. Whisper, with more parallelizable branches, improves further, from 2,804.7 ms to 1,861.7 ms with 4 threads. After reaching maximum parallelization, Conformer and MobileViT continue to benefit slightly due to task stealing, which efficiently utilizes idle threads. Beyond 6 threads, improvements taper off, and slight increases occur, such as MobileViT-S rising to 229.7 ms at 7 threads, due to multithreading overheads like context switching. These results demonstrate that while our method effectively uses parallelization, there's a trade-off with multithreading overhead. Nonetheless, the latency reductions remain significant, underscoring the effectiveness of BAP in multithreaded environments.

## 5 CONCLUSION AND FUTURE WORK

We introduce BAP, a system that accelerates inference on mobile devices for ASR and transformer models. By employing CPU parallel execution, branch-aware memory allocation, and efficient multithreading management, BAP fully leverages CPU resources to reduce latency. BAP achieved latency reductions of 18% to 38% across various models, averaging 27% to 30%, with maximum improvements up to 38.5%. Memory allocation was reduced by an average of $4.1\times$ to $8.6\times$, with maximum savings up to $15.6\times$ compared to TFLite's naive plan. BAP also reduced energy consumption by up to 24.64% on high-end devices due to faster inference. These results highlight BAP's effectiveness in enhancing real-time inference on mobile devices.

While BAP demonstrates strong performance improvements in inference latency and energy efficiency, there are areas for further exploration. Our method focuses on CPU-based optimization, but future work could extend this approach to heterogeneous computing environments, such as GPUs and NPUs, as support for dynamic tensors becomes more widely available. Additionally, future research could investigate adaptive task scheduling and dynamic workload balancing to further optimize both energy consumption and performance across different edge devices.

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
