# OpenReview forum: "BAP: BRANCH-AWARE PARALLEL EXECUTION FOR FASTER DNN INFERENCE ON MOBILE CPUS"
_ICLR.cc/2025/Conference — ICLR 2025 Conference Withdrawn Submission_

### Official Review · Reviewer_XcJw · 2024-10-22

**Soundness:** 2
**Presentation:** 2
**Contribution:** 2
**Rating:** 5
**Confidence:** 3

**Summary:**

Deep neural networks are widely deployed on edge devices like mobile phones to serve the users on many different tasks, thus it is important to reduce its latency and engergy consumption. Existing works focus on sequential execution of the layers on mobile devices. The authors of this paper propose to parallelize independent branches of a model to acceleate the execution. Experiments show that the proposed method could achieve up to 38.5% speedup.

**Strengths:**

1. Enable the parallel execution of independent branches of deep nerual networks on mobile devices.
2. Experiments show that the proposed framework could achieve up to 38.5% speedup compard with the widely used inference library tflite on mobile platforms.

**Weaknesses:**

1. Scientific novelty is limited, this contribution of this work is more on the engineering side.
2. More ablation study is needed.

See below for more details.

**Questions:**

1. Scientific novelty is limited, this work is more like a good engineering work.
There are a bunch of works [1, 2, 3, 4] that have studied the parallel execution of the independent branches of deep nerual networks on CPUs or GPUs. The schedule (how to group the branches for execution) proposed by this paper is a kind of greedy schedule that also has been widely studied. Thus, the contribution of this work lies more on the enginnering side from my perspective. Could you provide more comparison with the schedule used in this paper vs. the schedules used in prior works?

- [1] Scheduling Computation Graphs of Deep Learning Models on Manycore CPUs
- [2] Rammer: Enabling Holistic Deep Learning Compiler Optimizations with rTasks
- [3] IOS: Inter-Operator Scheduler for CNN Acceleration
- [4] Nimble: Lightweight and Parallel GPU Task Scheduling for Deep Learning

2. More ablation study is needed.
Since each single operator (like matrix multiplication and convolution) can also be parallelized among the mobile CPU cores. More abation study is recommended to show that the intra-operator parallelism is not enough to fully utilize the mobile CPU cores. From the Table 3, we can see that BAP can achieve 2~3x speedup when there are many branches in the model. Could you provide more profiling results to show that why the sequential execution can not saturate the mobile CPUs? Potential reasons are: 1) the workloads are too small for the powerful mobile CPUs, 2) the kernel implementation in sequential execution is not efficient to use multiple cores, 3) the sequential execution is bottlenecked by some resources (like DRAM bandwidth, computation units, etc) and parallelizing branches would alliviate this problem.

---

> ### Author Response · Authors · 2024-11-18
>
> `Q1:`  Thank you for the comment. We thoroughly reviewed the suggested works and acknowledge their contributions to scheduling and optimization. While these approaches are effective for static models, they do not address the dynamic properties we focus on, such as runtime-determined control flows and tensor sizes in transformer-based and ASR models.
>
> The key challenges with dynamic models are twofold:
> - **Incompatibility with Static Scheduling:** Pre-determined schedules fail when tensor shapes or control flows change at runtime, often leading to crashes.
> - **Impact of Memory Reallocation:** Frequent memory reallocations disrupt static scheduling plans, limiting parallelization feasibility.
>
> With the rise of transformer-based architectures, these issues are becoming increasingly common. Our work addresses this gap by reorganizing memory allocation to reduce contention and enable safe parallel execution. While our execution scheduling remains straightforward, the focus is on tackling the critical research gap left by SOTA frameworks like TFLite, MNN, and PyTorch Micro, which do not handle these dynamic properties [1, 2]. We believe this direction is valuable for inspiring future research on accelerating models with dynamic workloads and guiding the design of future frameworks or mobile-compatible models.
>
> [1] TensorFlow GitHub Issue #48553: [https://github.com/tensorflow/tensorflow/issues/48553](https://github.com/tensorflow/tensorflow/issues/48553)
> [2] Arm NN GitHub Issue #773: [https://github.com/ARM-software/armnn/issues/773](https://github.com/ARM-software/armnn/issues/773)
>
> `Q2:` Thank you for the insightful suggestion. In our experiments, the TFLite baseline uses intra-operator parallelization, with the number of threads explicitly set through the API `TfLiteStatus SetNumThreads(int num_threads);`. Despite this, our results show that branch-wise parallelism in BAP achieves significantly better latency improvements compared to TFLite's intra-operator parallelization, as shown in Table 3.
>
> A relevant discussion in TensorFlow GitHub Issue #20187 [1] highlights that the current intra-operator parallelism implementation often provides limited benefits. Moreover, intra-operator parallelism is insufficient for models with complex branching structures, as it focuses on individual operator acceleration without leveraging broader graph-level parallelism. These limitations make branch-wise parallelism a more effective and widely applicable solution.
>
> [1] TensorFlow GitHub Issue #20187: [https://github.com/tensorflow/tensorflow/issues/20187](https://github.com/tensorflow/tensorflow/issues/20187)

---

> ### Author Response · Authors · 2024-11-29
> **Follow-up**
>
> Dear Reviewer XcJw,
>
> As the discussion period is coming to an end, we would like to follow up to see if there are any remaining concerns or questions we can clarify. We have thoroughly addressed your feedback, highlighting the limitations of static scheduling for dynamic models, the challenges of runtime-determined control flows and the inefficiencies of intra-operator parallelism, which reflects the contribution and novelty of our work. These points are supported by updated results and discussions, emphasizing BAP’s effectiveness in addressing these critical gaps.
>
> If there are no further questions, we kindly ask you to consider updating the score to reflect these clarifications and the contributions of our work. Please let us know if further discussion would be beneficial.
>
> Best regards,
>
> Authors

---

> ### Author Response · Authors · 2024-12-03
> **Follow-up 2**
>
> Dear Reviewer XcJw,
>
> Today is the final day for score updates. We have carefully responded to your comments in the rebuttal and hope you can review it and consider increasing our score.
>
> Thank you for your time and effort.
>
> Authors

---

### Official Review · Reviewer_MhbN · 2024-10-30

**Soundness:** 4
**Presentation:** 4
**Contribution:** 3
**Rating:** 6
**Confidence:** 4

**Summary:**

The authors propose BAP that aims at improving runtime performance of DNN models on CPU devices. They achieve this by categorizing nodes in the DNN graph into sequential nodes, which can be grouped into a set for parallelism, and branch nodes, which defines the "set boundaries". The sets are then top-sorted to resolve dependencies. The authors also considered memory allocation and thread management during parallel execution to resolve potential memory conflicts and minimize multi-threading overhead. They evaluated their proposed algorithms with several open-source ASR and image recognition models on several hardware. For the metrics they are concerned (latency, memory usage, power), the proposed algorithm outperforms their chosen baselines in several aspects.

**Strengths:**

The paper is well-written and easy to follow. References to figures and tables are adequate. The idea is moderately novel: increase parallelism in model inference for CPU execution.  The empirical results, though lacking in some aspects, are supportive of their hypothesis. However, I am less confidence about its relevance to the industry as devices with GPUs become more prevalent. As models gets larger, it's hard to imagine running them without GPUs. Smaller models, according to the paper, show less gain with or without parallelism on CPUs.

**Weaknesses:**

The evaluation using only 10 small images and 5 short audio samples repeated five run is a bit inadequate. I'd encourage the authors to also include long audio samples too. Based on the argument of the paper, longer audios should even exemplify the advantage of the algorithm.
The authors can also improve the analysis in Section 4.3.3. by developing  relationships between the percentage of sequential nodes and the various performance metrics. Is it expected that higher percentage of sequential nodes lead to better performance? This could also lead to model design that is targeted for CPU devices.
Also please share the code base if possible.

**Questions:**

While running large models on CPU is an option, I wonder whether it is a viable option, hence its relevance. Many high-end devices come with GPU-like units nowadays, what's the scenario when we need to fall back to CPU? Maybe smaller models that do not wish to compete with large models, like a wake up detector? In that case, are there any other model inference acceleration targeting CPU that should be compared with?
Also how does the algorithm compare with GPU/TPU execution? Can it be a reliable fallback when GPUs get too busy?
Another aspect I'm interested in is how many other processes there were running on the CPUs when the measurements are taken. Does existing CPU usage affect the usage in any way?

---

> ### Author Response · Authors · 2024-11-18
>
> `W1:` Thank you for the detailed feedback.
>
> **Longer Audio Samples:**
> We conducted 15-second audio testing on a Google Pixel device to evaluate our approach. The results demonstrate up to 42% latency improvement in some cases, with average improvements shown below:
>
> | Model                     | Latency Ours (ms) | Latency TFLite (ms) | Improvement |
> |---------------------------|-------------------|---------------------|-------------|
> | Conformer CTC-15second    | 203.523          | 272.346             | 25.27%      |
> | Whisper-15second          | 1902.899         | 2716.482            | 29.95%      |
> | Conformer CTC-3to10second | 159.3224         | 211.6586            | 24.73%      |
> | Whisper-3to10second       | 1872.074         | 2666.281            | 29.79%      |
>
> From the table, we observe that while absolute latency improvements vary, relative improvements remain consistent across different audio durations. This stability arises because longer audio inputs are subsampled into smaller segments (e.g., 10 or 15 seconds) to accommodate device memory constraints. Transformer-based models like Whisper and Conformer require significant memory for query-key matrices, making very long sequences infeasible on mobile devices.
>
> **Impact of Sequential Nodes:**
> We appreciate the suggestion to analyze the relationship between sequential nodes and performance metrics. Table 1 already shows the number of parallelized branches, reflecting the proportion of sequential and parallelizable layers. Additionally, as mentioned in our response to Reviewer rvAe, the percentage of parallelizable operators varies by model: MobileViT (small or XS) achieves 17.93%, Whisper 61.43%, and Conformer 8.46%. These percentages have been added to Table 1 for clarity.
>
> **Code Availability:**
> We plan to release our code base after publication to ensure reproducibility and facilitate further research.
>
> `Q1: ` Thank you for the insightful questions.
>
> **Justification of CPU Execution:**
> While many devices include GPU-like units, dynamic-sized tensors and control flow, as seen in transformer-based and ASR models, cannot be accelerated using current delegation methods and default to CPU execution. This limitation is discussed in TensorFlow GitHub Issue #48553 [1]. These dynamic properties, particularly prevalent in token-by-token predictions, make CPUs the most reliable option. Additionally, CPU execution remains critical for many IoT devices.
>
> **Comparison with Other Baselines:**
> Methods targeting CPU inference acceleration, such as TFLite's memory optimization tool [2], Arm NN [3], and Qualcomm's delegation work [4], are designed for models without dynamic control flows. Consequently, they cannot address the complexities of models like Whisper or ASR systems (as discussed in our responses to Reviewer rvAe Weakness 3 and 1oc7 Question 6). Using TFLite's default CPU runtime ensures a robust and fair baseline, given its widespread usage and reliability.
>
> **Impact of Existing CPU Usage:**
> In our experiments, only the testing application was running to ensure consistent measurements. We acknowledge that existing CPU workloads can impact inference performance. Addressing this would require further research to handle resource contention dynamically, and enhancing BAP for multi-task scenarios is part of our ongoing work.
>
> [1] TensorFlow GitHub Issue #48553: [https://github.com/tensorflow/tensorflow/issues/48553](https://github.com/tensorflow/tensorflow/issues/48553)
> [2] Nicholas D. Lane. "Neural networks on microcontrollers: saving memory at inference via operator reordering." arXiv preprint arXiv:1910.05110 (2019). [https://github.com/eliberis/tflite-tools](https://github.com/eliberis/tflite-tools)
> [3] Arm NN: [https://github.com/ARM-software/armnn](https://github.com/ARM-software/armnn)
> [4] Qualcomm Blog: [https://www.qualcomm.com/developer/blog/2020/11/tensorflow-lite-inference-edge](https://www.qualcomm.com/developer/blog/2020/11/tensorflow-lite-inference-edge)

---

> ### Author Response · Authors · 2024-11-29
> **Follow-up**
>
> Dear Reviewer MhbN,
>
> As the discussion period nears its conclusion, we wanted to follow up to check if there are any additional points or concerns we can address. We have provided detailed responses to your feedback, including latency benchmarks for longer audio samples, an analysis of sequential versus parallelizable nodes, and clarified the justification for CPU execution and baseline comparisons. These updates, along with the percentages of parallelizable operators and their implications on performance, are reflected in the revised manuscript and updated tables.
>
> If there are no further questions, we kindly ask you to consider increasing the score based on the improvements and clarifications provided. Please let us know if any further discussion is needed.
>
> Best regards,
>
> Authors

---

> ### Author Response · Authors · 2024-12-03
> **Follow-up 2**
>
> Dear Reviewer MhbN,
>
> Today is the final day for score updates. We have carefully responded to your comments in the rebuttal and hope you can review it and consider increasing our score.
>
> Thank you for your time and effort.
>
> Authors

---

### Official Review · Reviewer_1oc7 · 2024-11-03

**Soundness:** 2
**Presentation:** 2
**Contribution:** 2
**Rating:** 5
**Confidence:** 3

**Summary:**

* The paper examines the challenges of executing models that require dynamic control flows on specialized processors and presents optimizations for executing these models on CPUs for edge devices.
* The paper proposes a branch-aware memory allocation policy (BAP), which improves memory utilization and facilitates efficient parallelization of the model, reducing the latency of DNN models.
* The paper presents results demonstrating the effectiveness of the BAP algorithm.

**Strengths:**

* The paper addresses a significant challenge for edge devices and employs a practical approach for implementing the solution, using TFLite as the foundational code. This choice enhances the potential for widespread adoption of the solution.

**Weaknesses:**

* Although the paper addresses an important problem and includes testing across multiple devices, some sections require rewriting to clearly highlight the contributions and provide sufficient supporting evidence. The major contributions of the paper, as presented in lines 85-91 (with inlined feedback), are:

> * We introduce BAP, a CPU-specific optimization system designed for ASR and transformer-based models with dynamic control flows and tensor operations.
>> It is not clear if this point includes the other two points. Does this refer to multithreading optimization? It is not clear, and I recommend rephrasing this to present the contribution more explicitly.

> * We develop a novel branch-aware memory allocation strategy that minimizes memory conflicts and optimizes cache usage by isolating memory arenas for parallel branches.
>> * The term "memory conflict" is unclear and is not discussed in detail within the paper. Additionally, there is no result presenting evidence related to cache optimization.

> * BAP significantly reduces latency, memory usage, and energy consumption without requiring model refactoring, making it practical for real-time inference on mobile edge devices.
>> * This point overlaps significantly with the above points. Please rephrase the contributions to more clearly differentiate the primary contributions.

* The paper can benefit by improving formal sections of the paper. E.g.
  * Line 176-179: "A branch b, or subgraph S, is a set of sequentially connected nodes { v1,v2,...,vk} within a layer.  A subgraph may have multiple inputs or outputs, but within the subgraph, the nodes are strictly connected in a sequential manner. This ensures that each subgraph can be executed independently of others within the same layer, provided there are no inter-branch dependencies."
  > * Is the term branch synonymous with subgraph? If so, using a consistent term could enhance readability.
  > * Last part "provided there are no inter-branch dependencies." looks redundant by the definition, in case of misunderstanding, please describe the cases it covers.
  * In algorithm 1, line 15-24, please add a condition when a node is marked as visited.

* For the results section, it will be helpful to provide more details about baseline. There are many implementations of vendor-specific Tflite  delegates e.g. [1] [2] and other frameworks for CPU execution. Kindly add a strong justification of using TFlite default implementation as the baseline. Further,
> Line 354-358: "While we compared allocation memory efficiency against both plans, we focused on comparing other aspects like latency and energy consumption against the first, as it represents TFLite’s optimal performance.  Also, we did not include methods like CoDL or NN-Stretch, as they mainly focus on DNNs without dynamic control flows and rely on heterogeneous processor co-execution. In this case, the TFLite runtime remains the SOTA for CPU-only execution."
>> It is not clear TFLite runtime as been described as SOTA for "allocation memory efficiency" or latency. Please provide an appropriate reference for the same.

* There is a lack of clarity on the TFlite's capability that is considered as baseline based on following lines.
Lines 375-377: Although TFLite also utilizes multi-core processing, its multithreading support for floating-point models remains limited. Specifically, TFLite’s parallelization is constrained to certain operations, leaving much of the computation serialized and diminishing its ability to fully capitalize on multi-core architectures"
> Does the TFLite implementation used in the experiment support multithreading? This is crucial for understanding whether the observed latency improvement should be attributed to "parallel vs sequential execution" or to enhanced parallelization due to the BAP algorithm.

[1] https://github.com/ARM-software/armnn

[2] https://www.qualcomm.com/developer/blog/2020/11/tensorflow-lite-inference-edge

**Questions:**

Please review my comments in "weakness" section. The same may be due to lack of understanding.

---

> ### Author Response · Authors · 2024-11-18
>
> `W1:` Thank you for the suggestions. To address the overlap and ensure clarity, we already rephrased the contributions as:
>
> 1. "We introduce BAP, a CPU-specific optimization system for ASR and transformer-based models with dynamic control flows and tensor operations. BAP combines branch-aware memory allocation and multithreading to optimize parallel execution and reduce latency."
>
> 2. "We develop a branch-aware memory allocation strategy that reduces data contention by isolating memory arenas for parallel branches, enabling safe and efficient parallel execution."
>
> 3. "BAP delivers substantial improvements in latency, memory efficiency, and energy consumption without requiring model refactoring, supporting practical real-time inference on edge devices"
>
> `W2-1:` Thank you for the suggestion. Regarding the term "branch," it is synonymous with "subgraph" in our context. To enhance readability, we will ensure consistent terminology by using "branch" throughout the paper. Regarding the phrase "provided there are no inter-branch dependencies," we agree it may be redundant given the earlier definition and will streamline this part for clarity.
>
> `W2-2:` Thank you for pointing this out. After **Line 15**, we have added the following lines (See the updated Algorithm 1):
>
> ```plaintext
> 16:     if \(v_i\) is visited then
> 17:         continue
> 18:     end if
> ```
>
> `W3-1:`  Thank you for the question and for highlighting the works of Arm NN and Qualcomm. We carefully reviewed their repositories and implementations to address this point. As mentioned in our paper, many proposed methods, including vendor-specific delegates, do not handle dynamic-sized tensors or control flow operations effectively. This limitation is particularly relevant with the increasing deployment of transformer-based and ASR models, which frequently exhibit these characteristics on mobile devices.
>
> In the Arm NN GitHub issue #773 [1], this limitation is explicitly discussed, where the Whisper TFLite model fails with Arm NN due to its reliance on dynamic-sized tensors. Additionally, TFLite is one of the most widely adopted frameworks for embedded and mobile systems, with deployment on over 4 billion devices globally [2]. This further supports our choice to use TFLite’s default CPU implementation as the baseline, as it remains the most reliable option for such models.
>
> [1] Arm NN GitHub Issue #773: [https://github.com/ARM-software/armnn/issues/773](https://github.com/ARM-software/armnn/issues/773)
> [2] TensorFlow Lite – Computer Vision on Edge Devices [2024 Guide]: [https://capalearning.com/2024/10/21/tensorflow-lite-computer-vision-on-edge-devices-2024-guide/](https://capalearning.com/2024/10/21/tensorflow-lite-computer-vision-on-edge-devices-2024-guide/)
>
> `W3-2:` Thank you for the question. TFLite’s arena-based memory allocation strategy is widely recognized for its efficiency in handling resource-constrained environments using greedy heuristics, as noted in [1]. Regarding latency, TFLite integrates multi-threading and optimized operators for CPU execution, making it a strong baseline for CPU-only performance, as evidenced in TensorFlow's official documentation and benchmarks [2]. We will clarify these points in the revised manuscript.
>
> [1] TelaMalloc: Efficient On-Chip Memory Allocation for Production Machine Learning Accelerators, Martin Maas et al.
> [2] TensorFlow blog, "Optimizing TensorFlow Lite runtime," [https://blog.tensorflow.org/2020/10/optimizing-tensorflow-lite-runtime.html](https://blog.tensorflow.org/2020/10/optimizing-tensorflow-lite-runtime.html).
>
> `W4:` Thank you for the question. The TFLite implementation in our experiments supports multithreading, which we enabled using the `TfLiteStatus SetNumThreads(int num_threads)` API to fully utilize all available CPU cores, ensuring a fair comparison with BAP. The improvements observed in our paper are attributed to BAP’s graph-level parallelization. We will clarify this in the revised manuscript.

---

> ### Author Response · Authors · 2024-11-29
> **Follow-up**
>
> Dear Reviewer 1oc7,
>
> As the discussion period nears its conclusion, we would like to ask if there are any remaining questions or concerns we can address. We have carefully incorporated your feedback, refining our contributions, terminology, and algorithm details, as well as clarifying key aspects like our baselines and multithreading comparisons with TFLite.
>
> If there are no further points of clarification, we respectfully ask Reviewer 1oc7 to consider increasing the score based on the addressed concerns and improvements. Please let us know if further discussion would be helpful.
>
> Best regards,
>
> Authors

---

> ### Author Response · Authors · 2024-12-03
> **Follow-up 2**
>
> Dear Reviewer 1oc7,
>
> Today is the final day for score updates. We have carefully responded to your comments in the rebuttal and hope you can review it and consider increasing our score.
>
> Thank you for your time and effort.
>
> Authors

---

### Official Review · Reviewer_rvAe · 2024-11-04

**Soundness:** 3
**Presentation:** 2
**Contribution:** 3
**Rating:** 5
**Confidence:** 3

**Summary:**

This paper presents BAP, a method to enhance neural network inference on mobile CPUs for models with multi-branch structures. BAP implements a branch-aware memory allocation strategy that isolates memory arenas for parallel branches, thereby improving cache locality and reducing memory overhead. Evaluated on ASR and transformer-based models, BAP achieves up to 38.5% reduction in inference latency, decreases memory allocation requirements by as much as 15.6×, and cuts energy consumption by up to 20.2% compared to the TFLite default memory allocator.

**Strengths:**

+ This work identifies the potential of parallel execution to improve the performance of models with multi-branch structures.
+ The authors address critical challenges, such as dynamic allocation and operator reordering, which arise with parallel execution.
+ Multiple evaluation metrics—latency, memory usage, and power consumption—are used to comprehensively demonstrate BAP's benefits.

**Weaknesses:**

- Adding figures to illustrate the multi-branch and dynamic properties of existing neural networks would improve clarity.
- Providing an overview of CPU and GPU computing capabilities would help emphasize the need for CPU-specific optimizations, or clarify why other processors are less suited for the targeted issues.
- The comparison is limited to TFLite; incorporating recent benchmarks such as operator reordering [1] would provide a stronger baseline.
[1] Liberis, Edgar, and Nicholas D. Lane. "Neural networks on microcontrollers: saving memory at inference via operator reordering." arXiv preprint arXiv:1910.05110 (2019).

**Questions:**

The work would benefit from additional data and explanations to clarify its contributions and significance. Key questions and concerns include:
1. How does BAP improve cache locality, as mentioned in the abstract?
2. What percentage of operators, or computational workload, can be parallelized effectively by this approach?
3. What is the quantified benefit of parallel versus sequential execution across different models?
4. How does BAP address workload imbalance among branches? This explanation could be refined.
5. Many mobile CPUs now use asymmetric core designs—has this hardware feature been considered in the proposed solution?

---

> ### Author Response · Authors · 2024-11-18
>
> `W1:` We appreciate the suggestion and have enhanced Figure 1 to illustrate how runtime uncertainties in tensor sizes necessitate frequent memory reallocations, disrupting branch-level isolation. See the updated paper.
>
> `W2:` We appreciate the suggestion and clarify that CPU-specific optimizations are addressed in the Introduction and Related Work sections:
>
> - **CPU and GPU Performance:** As mentioned, on mobile devices, the performance gap between CPUs and GPUs for inference tasks is minimal, making CPUs a flexible and viable choice (See citations in our paper Zhang et al., 2023; Bordawekar et al., 2010).
>
> - **Dynamic Tensor Limitations:** Hardware-specific delegates (e.g., GPUs, NPUs) often fail with dynamic tensor sizes in ASR and transformer models, leading computations to fall back to the CPU (TensorFlow GitHub Issue #48553 [1]).
>
> - **IoT and Edge Devices:** CPUs dominate IoT and embedded systems. The embedded processor market is projected to grow from USD 19.36 billion (2019) to USD 32.53 billion (2028) [2], while the IoT market is expected to grow from USD 98.06 billion to USD 336.64 billion within five years [3], with many devices relying solely on CPUs for cost-effective, power-efficient applications.
>
> 1. TensorFlow GitHub Issue #48553: [https://github.com/tensorflow/tensorflow/issues/48553](https://github.com/tensorflow/tensorflow/issues/48553)
> 2. Allied Market Research: [https://www.alliedmarketresearch.com/embedded-processor-market](https://www.alliedmarketresearch.com/embedded-processor-market)
> 3. Mordor Intelligence: [https://www.mordorintelligence.com/industry-reports/iot-devices-market](https://www.mordorintelligence.com/industry-reports/iot-devices-market)
>
> `W3:` We appreciate the suggestion to incorporate the operator reordering benchmark tool. However, after attempting to apply it to models, we found several limitations:
>
> - **Unsupported Operations:** The tool does not support all operations in our models, leading to errors (e.g., `KeyError`).
> - **Single Output Limitation:** The tool assumes single-output operations, making it incompatible with our models, which involve multi-branch architectures and multiple output nodes (e.g., `AssertionError`).
>
> Additionally, our work focuses on accelerating inference through branch-aware memory allocation, not on memory optimization. By isolating parallel branches, we minimize interference and enable efficient multithreading. As shown in Table 2, while our memory usage is slightly higher than TFLite’s optimized runtime, this trade-off achieves faster inference and higher energy efficiency for models with dynamic control flows.
>
> `Q1:` BAP improves cache performance by isolating memory arenas for parallel branches, reducing contention and ensuring efficient memory reuse. We acknowledge that "cache locality" may not fully capture this mechanism. The abstract has been refined to:
> "...reducing contention and optimizing memory reuse within each branch."
>
> `Q2:` We thank the reviewer for this insightful question. The percentage of parallelized operators varies by model: MobileViT (small or XS) achieves 17.93%, Whisper 61.43%, and Conformer 8.46%. Table 1 already shows the number of parallelized layers, and we have updated it to include these percentages for clarity.
>
> `Q3:` Thank you for the question. The quantified benefit is shown in Figure 2, demonstrating overall latency improvements, and in Table 3, where parallelizable layers like Layer 1 in Whisper (67.7%) and Layer 2 in MobileViT-XS (58.1%) achieve significant reductions. These results highlight the efficiency of parallel execution in multi-branch layers, and the updated Table 1 includes these values.
>
> `Q4:` Thank you for your suggestion. To clarify how BAP addresses workload imbalance, we will revise the explanation to describe the process in two steps:
>
> - **Workload Quantification:** During graph structure analysis, we approximate each branch's workload by counting operations. Lightweight operations (e.g., reshape) are assigned a weight of 0.5, while heavier operations (e.g., dense layers, convolutions) are assigned a weight of 1. This provides a quantified workload for scheduling.
>
> - **Scheduling:** Based on the quantified workload, branches with balanced workloads are grouped for parallel execution. Heavier or lighter branches are scheduled sequentially, either before or after the parallel branches, depending on their position in the computation graph. This approach ensures efficient execution without bottlenecks.
>
> `Q5:` Thank you for highlighting asymmetric core designs in mobile CPUs. While BAP does not currently account for core heterogeneity, future work could assign tasks to core types (e.g., big cores for heavy workloads, LITTLE cores for light ones) to further optimize energy and performance. We appreciate this insightful suggestion and plan to explore this avenue in future iterations of our method.

---

> ### Author Response · Authors · 2024-11-29
> **Follow-up**
>
> Dear Reviewer rvAe,
>
> As the discussion period is nearing its end, we wanted to check if there are any follow-up points requiring clarification kindly. We have thoroughly addressed all raised concerns in our response, including enhancing Figure 1, incorporating additional benchmarks and data, and refining key explanations throughout the manuscript. Furthermore, we conducted extensive experiments and updates, such as detailing parallelized operator percentages, latency improvements, and workload scheduling mechanisms, as reflected in Tables 1–3.
>
> If there are no further questions or concerns regarding these updates, we would respectfully ask Reviewer rvAe to consider increasing the score to reflect these clarifications and improvements. Please let us know if further discussion would be helpful.
>
> Best regards,
>
> Authors

---

> > ### Author Response · Authors · 2024-12-03
> > **Follow-up 2**
> >
> > Dear Reviewer rvAe,
> >
> > Today is the final day for score updates. We have carefully responded to your comments in the rebuttal and hope you can review it and consider increasing our score.
> >
> > Thank you for your time and effort.
> >
> > Authors

---

### Author Response · Authors · 2024-11-19

Dear Reviewers and Meta-Reviewers,

Thank you for your valuable feedback and the engaging discussions on our submission. We have carefully addressed all concerns raised and provided detailed clarifications and additional experiments in our rebuttal.

We also ran the reviewer-suggested baseline codes but found that they could not adapt to the dynamic properties of ASR and Transformer models, which is precisely the challenge our work addresses. This aligns with our motivation and highlights the importance of tackling these issues for practical deployment.

We deeply appreciate the alignment of many reviewer comments with the challenges we encountered during this work and found the exchange of ideas both enriching and motivating. We hope to continue these discussions and are eager to hear your thoughts on our responses.

Thank you for revisiting our rebuttal, and we look forward to your feedback!

Best regards,

Authors

---

### Author Response · Authors · 2024-12-02
**Friendly Reminder: Reviewer Responses for #9257 on OpenReview**

Dear Reviewers and Meta-Reviewers,

I hope this message finds you well. I wanted to kindly remind you that today marks the final day for posting reviewer responses on OpenReview for our paper.

We have carefully and actively addressed all your feedback and concerns and worked to improve the clarity and quality of our work based on your comments. We hope our efforts reflect positively in your evaluation, including a possible score increase if you find our revisions satisfactory.

Thank you again for your valuable input. Please don’t hesitate to reach out if there’s anything else we can assist with.

Best regards,

Authors

---

### Author Response · Authors · 2024-12-03
**Final Day for Review Updates**

Dear Reviewers and Meta-Reviewers,

We deeply appreciate the time and effort you have dedicated to reviewing our paper. ``While there has been no interaction during this process, we sincerely hope that our efforts in addressing the review comments are recognized``.

If there is still an opportunity, we kindly ask you to consider updating our score today, the final day for changes. Your feedback and evaluation mean a great deal to us.

Thank you again for your consideration and support.

Best regards,

Authors

---

### Note · Authors · 2024-12-10

**Comment:**

Hi all,

We would like to formally withdraw our paper. We were disappointed by the lack of engagement during the rebuttal phase. This has led us to conclude that our work has not been given fair and thorough consideration, especially as we observed interactions for other submissions during this period.

We hope that future iterations of the review process will better support meaningful interactions between authors and reviewers.

Thank you for your time and effort in reviewing our submission.

Best regards,

Submission9257 Authors

**Withdrawal Confirmation:**

I have read and agree with the venue's withdrawal policy on behalf of myself and my co-authors.